# Sexual Satisfaction among Lebanese Adults: Scale Validation in Arabic and Correlates Following Two Cross-Sectional Studies

**DOI:** 10.3390/healthcare11233068

**Published:** 2023-11-30

**Authors:** Cedric Abboud, Mirna Fawaz, Rabih Hallit, Sahar Obeid, Diana Malaeb, Souheil Hallit

**Affiliations:** 1School of Medicine and Medical Sciences, Holy Spirit University of Kaslik, Jounieh P.O. Box 446, Lebanon; cedric.j.abboud@net.usek.edu.lb (C.A.); hallitrabih@hotmail.com (R.H.); 2Faculty of Health Sciences, Beirut Arab University, Tareek Al Jadida, Afeef Al Tiba, Beirut P.O. Box 11-5020, Lebanon; mirna.fawaz@bau.edu.lb; 3Department of Infectious Diseases, Bellevue Medical Center, Mansourieh P.O. Box 295, Lebanon; 4Department of Infectious Diseases, Notre Dame des Secours University Hospital, Byblos P.O. Box 3, Lebanon; 5Social and Education Sciences Department, School of Arts and Sciences, Lebanese American University, Byblos P.O. Box 36, Lebanon; saharobeid23@hotmail.com; 6College of Pharmacy, Gulf Medical University, Ajman P.O. Box 4184, United Arab Emirates; 7Applied Science Research Center, Applied Science Private University, Amman 11931, Jordan; 8Research Department, Psychiatric Hospital of the Cross, Jal Eddib P.O. Box 60096, Lebanon

**Keywords:** sexual satisfaction, waterpipe dependence, emotional intelligence, Lebanon

## Abstract

(1) Background: Sexual satisfaction (SS) is an essential component of quality of life. There is a scarcity of research about sexual satisfaction in Lebanon, a country where discussing sexual issues is still considered a taboo. The present study aimed to assess the reliability and validity of responses to the items in the Arabic version of the Sexual Satisfaction Questionnaire (SSQ), as well as the correlates of sexual satisfaction, among a sample of Lebanese adults. (2) Methods: Two cross-sectional studies were conducted between June and September 2022 with 270 and 359 participants, respectively. (3) Results: The results showed that the Sexual Satisfaction Questionnaire is adequate to be used in Lebanon (McDonald’s ω = 0.90 and 0.86, respectively). Multivariate analysis showed that higher waterpipe dependence (Beta = −0.17) was substantially linked to lower sexual satisfaction, while better emotional intelligence (EI) (Beta = 0.27) and physical activity (Beta = 0.17) were significantly associated with greater sexual satisfaction. (4) Conclusions: The reliability and validity of the responses to the Arabic version of the Sexual Satisfaction Questionnaire were supported by our findings. Also, practical implications for sexual satisfaction enhancement strategies in the Lebanese population might be beneficial since many associated factors are considered to be modifiable.

## 1. Introduction

Sexual satisfaction (SS) is an emotional reaction carried by a person’s evaluation of the different positive and negative characteristics of his/her sexual life [1]. The World Health Organization suggests a wide understanding of sexual health that incorporates pleasure, well-being, and satisfaction [2]. Thus, sexual satisfaction is a key aspect of someone’s sexual health, and also an essential component of quality of life for both men and women [3].

In order to assess sexual satisfaction according to the most recent comprehensive definition of sexual health, Nomejko, A. and Dolińska-Zygmunt, G. developed the Sexual Satisfaction Questionnaire (SSQ) in Poland, back in 2014. The responses to the items of this questionnaire have been validated and found to be reliable by the findings of numerous investigations for assessing someone’s attitude toward his/her own sexual activity (both cognitively and emotionally) among young, middle-aged, and even older adults [4]. Besides the psychometric characteristics of the responses to its items, the SSQ stands out as one of the few sexual satisfaction assessment techniques built on a well-defined holistic approach, considering the social, emotional, mental, and physical elements of sexual satisfaction [5]. Additionally, completing the SSQ only takes a few minutes, making it an extremely time-effective test. An official English translation was provided by the authors along with the original version in Polish. Moreover, this scale has been translated into other languages such as Italian [6] and Bengali [7]. Given the unique socio-cultural context of the Arab world, where conservative attitudes towards sexuality often prevail, the availability of a validated Arabic version of the questionnaire can facilitate a more nuanced understanding of the intricate dynamics that potentially influence sexual satisfaction in this cultural setting. This version can account for specific cultural factors that may impact sexual behaviors, attitudes, and perceptions, thereby allowing researchers and clinicians to obtain comprehensive and accurate insights into the sexual well-being of Arabic-speaking individuals. Furthermore, an Arabic version of the questionnaire can foster inclusivity and accessibility in research and clinical practice, ensuring that the assessment of sexual satisfaction is not hindered by language barriers or cultural insensitivity.

Unfortunately, due to social norms and this subject being considered as a taboo topic in many countries including Lebanon, especially among the older population, patients avoid talking about sexual satisfaction with their health care provider [8]. Also, physicians do not discuss sexual issues frequently with their patients. For instance, a previous survey involving Lebanese obstetrics and gynecology physicians discovered that only 31% of them almost always questioned their patients about their sexual health [9]. Therefore, it may not be surprising that despite the evolving global discourse on sexual health and rights, there exists a noticeable scientific gap within the Lebanese society. Studies highlight the misconceptions and inadequate knowledge, especially among Lebanese youth regarding reproductive health, contraception, and sexually transmitted infections [10,11]. Limited access to comprehensive sexual education and healthcare services further exacerbates the challenges faced by young individuals in navigating their sexual health. Therefore, there is an urgent need for more research to fill this gap.

Previous investigations into the factors behind sexual satisfaction have provided notable findings, starting with some unhealthy lifestyles like waterpipe dependence. To the best of our knowledge, the potential effect of waterpipe dependence on sexual satisfaction has not been studied yet, although individuals of both genders in the Middle East and North Africa (MENA) area commonly smoke tobacco through waterpipes, also referred to as narghiles, hookahs, or shisha [12]. For instance, Lebanon has the highest prevalence of waterpipe smoking in the region with 36.9% [13]. In addition, there is little knowledge about waterpipes among the Lebanese population in general, as there is a widespread misconception that waterpipe smoking is safer than cigarettes [14]. However, it is actually known that waterpipe smoking is possibly associated with a number of deleterious general health outcomes, including heart diseases and cancer [15].

Furthermore, tobacco smoking is also one of the leading unhealthy lifestyles. Indeed, chronic smoking has been demonstrated to be a risk factor for erectile dysfunction in men, which can potentially and indirectly affect levels of sexual satisfaction and levels of desire [16] and cause delayed orgasm among females, leading to an overall deterioration in sexual partner relationships [17].

Another lifestyle factor is alcohol abuse. According to a recent review article based on prospective studies published between August 1954 and November 2020, a prevalence rate of 75.0% for sexual dysfunction was reported in individuals with alcohol use disorder [18], with rates being much higher in alcohol-dependent individuals than in social drinkers. This is why it was interesting to study its association with sexual satisfaction in our sample.

Belonging to the lifestyle category, higher levels of physical activity indicated better sexual desire, function, and satisfaction as found in a study of university students [19]. Moreover, men and women at risk for sexual dysfunction, regardless of physical activity level, were found to benefit from exercising more rigorously. Specifically, men exhibited improvements in erectile dysfunction, while women reported reduced dissatisfaction with orgasms and decreased difficulty in arousal, as found by a large study recruiting cyclists, runners, and swimmers in the United States, Canada, Great Britain, Australia, and New Zealand [20]. Yet, interestingly, a recent study showed that excessive physical activity was linked to poor sexual satisfaction among women under consultation for female sexual dysfunction in Italy [21]. However, given that most prior studies have indicated a positive correlation between physical activity and sexual satisfaction, we anticipated a similar pattern within our sample.

On the other hand, some sociodemographic characteristics like being married were also investigated in our research. In fact, single individuals were shown to have poorer sexual function compared to married couples in India [22]. In contrast, a prior study conducted among German adult subjects aged 30 and above found that marriage is not a determinant of sexual satisfaction [23].

Another sociodemographic characteristic is level of education. For instance, a cross-cultural study from four countries (Slovenia, Croatia, Bosnia and Herzegovina, and Romania) demonstrated higher levels of sexual satisfaction among individuals with a higher level of education [24]. Additionally, sexual satisfaction was much lower among patients with lower levels of education as found by a study of infertile men [25]. Based on the results of prior research, we expected a similar positive association between the level of education and sexual satisfaction.

Prior studies also largely investigated the gender differences regarding sexual satisfaction. For instance, two studies conducted in North America, each including 255 mixed-sex couples newly transitioning to parenthood, indicated that men are more satisfied sexually than women [26,27]. However, in late midlife as well as after retirement, women were found to have higher levels of sexual satisfaction than men [28].

In addition, a few studies have investigated psychological distress as a potential correlate and found that it has a negative association with sexual health, including sexual satisfaction in specific populations like infertile men as shown by a study conducted among 246 infertile couples in Jordan [29]. Also, psychological interventions were found to be effective treatment options for sexual dysfunction, therefore possibly ameliorating sexual satisfaction [30].

Moreover, the potential effect of other variables such as anxiety and depression on sexual health has been studied. For instance, in a Swedish study conducted in the general population, the presence of depression and anxiety symptoms had a significant association with sexual dysfunction, where “low sexual interest” was identified as the prevalent dysfunction in women and “low satisfaction with sex life” was indicated as the primary dysfunction in men [31]. Also, based on another study including heterosexual married couples from Turkey, a husband’s depression was negatively associated with both their own and the wives’ sexual satisfaction [32]. However, moderate levels of anxiety might facilitate sexual arousal leading to a higher level of sexual satisfaction, as found in a Spanish study among healthy young adults of both genders [33].

Lastly, emotional intelligence (EI) was found to have a significant positive correlation with sexual satisfaction after a comprehensive review of studies on PubMed and Google Scholar as shown, for example, by studies conducted among married women [34] and married men presenting at health centers in Iran [35]. Moreover, emotional intelligence was considered to be among the most important psychological variables related to the sexual satisfaction of the couple, and thus marital satisfaction in the USA [36]. It is also interesting to note that only one study conducted in Switzerland among men and women of all ages demonstrated a negative connection between EI and female sexual desire and sexual satisfaction [37]. However, since the majority of studies showed a positive relation between EI and sexual satisfaction, the current study will posit that there will be a positive relation between these two variables within a Lebanese sample of adults in the general population.

The present study’s aim was to assess the psychometric properties of the participants’ responses to the items of the Sexual Satisfaction Questionnaire, within a Lebanese adult population, thereby contributing to the validation of the questionnaire in this specific cultural context. As a secondary objective, this study investigated the various factors associated with sexual satisfaction in the Lebanese population. These findings would be crucial for suggesting future sexual satisfaction enhancement strategies that would focus on modifiable factors, hence possibly ameliorating the sexual health of the Lebanese population. We hypothesize that people with higher emotional intelligence, higher levels of physical activity, and lower waterpipe smoking would exhibit higher sexual satisfaction in our sample.

## 2. Materials and Methods

### 2.1. Study Design and Procedures

We conducted two cross-sectional studies between June and September 2022. All data were collected via Arabic questionnaires shared via Google Form links. The project was promoted on social media platforms and needed approximately 15–20 min to be completed. After confirming that all participants had given their informed consent digitally, the different instruments mentioned below were administered in a pre-randomized manner in order to reduce potential order effects. Participants in the research responded anonymously, willingly, and without any compensation [38].

The process of forward and backward translation has been used for the scales that are not validated in Arabic. The forward translation was initially completed by one bilingual translator who was native in Lebanese and fluent in English. A team of medical and linguistic experts approved the Arabic translation. The backward translation from Arabic to English was carried out by a native English speaker who was also proficient in Arabic, but without being familiar with the concepts of the scales. The experts group then compared the back-translated survey with the original English version. The procedure was carried out until all discrepancies and contradictions were eliminated.

### 2.2. Participants

All participants met the following “inclusion criteria”: having reached at least 18 years old, as well as being originally Lebanese.

### 2.3. Minimal Sample Size

The sample size was calculated using the G*Power software v. 3.0.10. Taking into consideration an alpha error of 5% and a power of 80%, a 10% baseline model R2, and a maximum number of 20 predictors that could be incorporated into the model, the minimum sample size needed was 205 participants [38].

### 2.4. Instruments

*The Sexual Satisfaction Questionnaire* (*SSQ*), a 10-item questionnaire, in which participants were required to describe their satisfaction with their sexual life using a 4-point Likert-type scale (with 1 being “not at all satisfied” and 4 “extremely satisfied”), was utilized. The items are divided into two categories: questions assessing positive emotions regarding sexual satisfaction (e.g., “I find my sexual life fulfilling”), and ones evaluating negative emotions (e.g., “My sexual life frustrates me”). Studies have supported the responses to this scale’s items as valid and reliable [4]. Total scores range from 10 to 40 with higher scores denoting higher sexual satisfaction [39].

Waterpipe Dependence was assessed with the *Lebanese Waterpipe Dependence Scale*-*11* (*LWDS*-*11*) [40]. This scale consists of 11 items, each of which is responded to on a 4-point Likert-type scale. An example of relevant items is the “Number of waterpipes smoked per week” with four corresponding response categories: 1–2; 3–4; 5–6; 7 and more. A score of 10 or greater was regarded as having high waterpipe dependence [41]. In the current study, the LWDS-11 items generated a McDonald’s omega value of ω = 0.77.

*The Fagerström Test for Nicotine Dependence* (*FTND*) was also used in this study. It includes 6 questions to assess the degree of nicotine dependence (ND) (e.g., “How many cigarettes do you smoke each day”). Higher scores reflect higher ND [42]. The classification of dependence was as follows: 0–2 (very low), 3–4 (low), 5 (moderate), 6–7 (high), and 8–10 (very high). This scale has been widely used and studied for the validity and reliability of the responses to its items in various clinical settings [43]. In the current study, the FTND items generated a McDonald’s omega value of ω = 0.76.

*The Alcohol Use Disorder Identification Test* (*AUDIT*) is a 10-item tool asking participants about their drinking habits, alcohol use, and problems related to alcohol (e.g., “How often during the last year have you failed to do what was normally expected from you because of drinking?”). Each response is graded from 0 (Never) to 4 (almost daily), with a maximum possible score of 40. Higher scores are linked to a higher risk of alcohol use disorder [44]. This scale is validated in Arabic [45]. In the current study, the AUDIT items generated a McDonald’s omega value of ω = 0.87.

*The Brief Emotional Intelligence Test* (*BEIS*-*10*) [46] is a 10-item self-reported questionnaire, assessing attitudes toward examining one’s personal and other people’s emotions (e.g., “I easily recognize my emotions as I experience them” and “I can tell how people are feeling by listening to the tone of their voice”). Each response is given a score on a Likert-type scale from 1 to 5, with 1 denoting “strong disagreement” and 5 expressing “strong agreement”. Higher scores indicate higher EI. Thus, a total score from 10 to 50 is calculated, with 50 being the highest emotional intelligence (EI) level possible. The construct validity and reliability of the responses to this scale have been previously investigated [47]. In the current study, the BEIS-10 items generated a McDonald’s omega value of ω = 0.87.

*The Sexual Dysfunction Questionnaire* (*SDQ*) was used in order to evaluate the convergent validity of the SSQ. The responses to the items in this questionnaire have been previously shown to be valid and reliable in order to briefly and easily measure sexual dysfunction. The questionnaire consists of 19 questions responded to on a 5-point Likert-type scale (ranging from Always, Frequently, Sometimes, Hardly Ever, to Never). Examples of relevant items include “sexuality scares me”, and “I avoid situations that arouse my sexuality”. Higher scores reflect more sexual dysfunction [48]. In the current study, the SDQ items generated a McDonald’s omega value of ω = 0.93.

*The Depression Anxiety Stress Scale 8* (*DASS*-*8*): This scale comprises eight items divided into three subscales: depression (e.g., felt down hearted and blue), anxiety (e.g., felt scared without reason), and stress (e.g., was using a lot of my mental energy). The ratings for each of the 8 items range from 0 (“did not apply to me at all”) to 3 (“applied to me the majority of the time”). Higher scores reflect more severe symptoms [49]. In the current study, the DASS-8 items generated a McDonald’s omega value of ω = 0.90.

Regarding demographics, participants were requested to provide their demographic information, including their age, gender, marital status, and degree of education. In order to calculate the Household Crowding Index that reflects the socioeconomic condition, we divided the total number of people by the overall number of rooms within the house, excluding the bathrooms and kitchen [50]. Participants were asked about their physical activity in terms of strength (1 = light activity to 5 = heavy breathing and constant sweating), frequency (1 = less than once a month to 5 = daily or almost daily), and duration (1 = less than 10 min to 4 = more than 30 min); those three values were multiplied together to create a physical activity index [51]. The self-reported weight and height were used to compute the Body Mass Index (BMI). Also, on a scale from 1 to 10, with 10 denoting extreme pressure, participants were asked to rate how much pressure they felt regarding their own financial burden in general.

### 2.5. Data Analysis

The dataset contained no missing answers. We employed an EFA-to-CFA methodology to investigate the factor structure of the Sexual Satisfaction Questionnaire [52]. Samples 1 and 2 were used, respectively, for the EFA and CFA.

We used the FACTOR software v. 12.04.02 to construct a principal-component EFA, applying the first sample in order to investigate the factor structure of the SSQ. The matrix was analyzed using the polychoric correlation and the promax rotation. We checked that every condition for item-communality [53], average item correlations, and item-total correlations was met [54]. The suitability of our sample was validated by the Kaiser–Meyer–Olkin (KMO) indicator of sampling adequacy (which should optimally be ≥0.80), and the Bartlett’s measurement of sphericity, which should be adequate [55]. The anti-image correlation matrix recommended that items with “fair” loadings and beyond (i.e., 0.40), and reasonable communality (i.e., 0.30) with low inter-item correlations (evocative of limited item redundancy), should be preserved [56].

Through applying the SPSS AMOS v.26 software and data of the second sample, we performed a CFA by using an estimation of the maximum likelihood. According to a prior study, a confirmatory or exploratory factor analysis requires a minimum sample size that is between three to twenty times the number of variables on the scale [57]. And based on the ratio of 20 respondents per item of the scale, which was surpassed in both samples, we estimated that a sample size of 200 participants was required to have sufficient statistical power. The maximum likelihood approach and fit indices were used to obtain parameter estimations. Also, this subsample’s average variance extracted (AVE) scores of ≥0.50 were deemed sufficient for assessing the convergent validity evidence [58].

Using the second sample, we performed a multi-group CFA to investigate the gender invariance of the SSQ scores [59]. All the metric, scalar, and configural levels of measurement invariance were evaluated [60]. According to the configural invariance, the structure of loadings of the latent variable(s) on indicators as well as the latent SSQ-10 variable(s) are comparable throughout all genders (that is, both sets of data should match the unconstrained latent model reasonably well). Through contrasting two nested models made up of a baseline model as well as an invariance one, the metric invariance was tested. The latter states that the size of the loadings is identical across genders. Finally, the same nested-model comparison technique was used to assess scalar invariance, which implies that item loadings and item intercepts are identical across genders [59]. We considered ΔCFI ≤ 0.010 and ΔRMSEA ≤ 0.015 or ΔSRMR ≤ 0.010 (0.030 for factorial invariance) as proof of invariance in accordance with the recommendations of Cheung and Rensvold [61] and Chen [59]. We intended to use an independent-samples *t*-test to assess for differences across genders on latent SSQ scores, only if partial scalar or scalar invariance were confirmed.

McDonald’s ω was used to evaluate the composite reliability for both subgroups, with scores higher than 0.70 indicating satisfactory composite reliability [62]. Given the documented issues with using Cronbach’s α as a measure of composite reliability, McDonald’s ω was chosen in this study [62]. Since the skewness and kurtosis values ranged from −1 to +1, the univariate normality of the Sexual Satisfaction Questionnaire was confirmed [55]. Student’s *t*-test was used to compare two means, whereas the Pearson test was used to correlate two continuous variables. Cohen (1992) stated that correlations with values of ≤0.10 were weak, those around 0.30 were moderate, and those equal to or greater than 0.50 were strong [63]. A linear regression was conducted, taking the sexual satisfaction score as the dependent variable. Factors that showed a *p* < 0.25 in the bivariate analysis were entered as independent variables in the final model.

## 3. Results

### 3.1. Description of the Sample

Table 1 illustrates detailed information regarding participants in both studies. Furthermore, 105 (29.2%) participants in study 2 showed high waterpipe dependence (LWDS-11 scores ≥ 10). Additionally, 302 (84.1%) had very low nicotine dependence, 18 (5.0%) low dependence, 7 (1.9%) moderate dependence, 19 (5.3%) high dependence, and 13 (3.6%) very high dependence, respectively.

### 3.2. Exploratory Factor Analysis (Sample 1)

#### 3.2.1. Factor Analysis

The 10 items of the SSQ had sufficient common variance for factor analysis according to Bartlett’s test of sphericity, χ^2^(45) = 2113.4, *p* < 0.001, and KMO (0.858). The EFA’s findings showed two factors (Factor 1: positive aspect of sexual satisfaction; Factor 2: negative aspect of sexual satisfaction) that accounted for 73.35% of the common variance. Table 2 reports all the factor loadings.

#### 3.2.2. Factor Structure Congruence and Composite Reliability

McDonald’s ω was acceptable for Factor 1 (ω = 0.86), Factor 2 (ω = 0.85), and the total score (ω = 0.90).

### 3.3. Confirmatory Factor Analysis (Sample 2)

CFA indicated that fit of the two-factor model of the SSQ acquired in the EFA was acceptable: χ2/df = 130.65/33 = 3.96, RMSEA = 0.091 (90% CI 0.075, 0.108), CFI = 0.941, TLI = 0.920. The factor loadings’ standardized values were all appropriate (check Table 2). As indicated by AVE = 0.52, the convergent validity of this model was reasonable. The correlation between the two factors was 0.52.

#### Composite Reliability

McDonald’s ω was acceptable for Factor 1 (ω = 0.80), Factor 2 (ω = 0.87), and the total score (ω = 0.86).

### 3.4. Gender Invariance (Sample 2)

All indices indicated that metric, scalar, and configural invariance were maintained across gender (Table 3). We computed an independent-samples *t*-test based on these findings to investigate gender variations in SSQ scores. Regarding sexual satisfaction, no discernible difference between men and women was discovered (24.70 ± 4.22 vs. 24.33 ± 3.66; *p* = 0.386; t(357) = 0.869; Cohen’s d = 0.093).

### 3.5. Convergent and Construct Validity (Sample 2)

Higher sexual dysfunction was significantly but weakly associated with less sexual satisfaction (r = −0.13; *p* = 0.016). More psychological distress (r = −0.17) was significantly but weakly correlated with less sexual satisfaction, whereas higher EI (r = 0.31) was significantly and moderately associated with more sexual satisfaction.

### 3.6. Bivariate Analysis (Sample 2)

The results of the bivariate analysis can be found in Table 4. A higher sexual satisfaction score was seen in married participants compared to single ones (26.22 ± 4.91 vs. 24.27 ± 3.70; *p* = 0.004; t(357) = 2.89, d = 0.45. No significant difference was found between participants who had a university education level vs. secondary or less (24.56 ± 3.87 vs. 23.00 ± 3.78; *p* = 0.066; t(357) = 1.84, d = 0.41) and between males and females (24.70 ± 4.22 vs. 24.33 ± 3.66; *p* = 0.386; t(357) = 0.87, d = 0.09). More waterpipe dependence, psychological distress, and alcohol use disorder were correlated with less sexual satisfaction, whereas higher physical activity index and older age were significantly associated with more sexual satisfaction (Table 4).

### 3.7. Multivariable Analysis (Sample 2)

Higher waterpipe dependence (Beta = −0.17) was significantly associated with less sexual satisfaction, whereas higher EI (Beta = 0.27) and physical activity (Beta = 0.17) were correlated with more sexual satisfaction (see Table 5).

## 4. Discussion

Starting with the validation of the Sexual Satisfaction Questionnaire, our findings provide greater support for the psychometric evidence of the participants’ responses to the SSQ, preliminarily suggesting the usefulness and feasibility of the Arabic form in assessing sexual satisfaction in Arabic-speaking people, at least in the Lebanese setting. We believe that the present findings would benefit researchers and clinicians in facilitating a more comprehensive discourse on sexual health in general while focusing on sexual satisfaction.

Using the CFA, our results confirmed the bi-dimensional structure of the SSQ, similar to the structure obtained in the original study [4]. The responses to the Arabic version of the Sexual Satisfaction Questionnaire have shown excellent psychometric characteristics and adequate internal consistency. The reliability coefficient that was determined by Cronbach’s α indicated a high consistency of 0.83 in previous research [29]. In our present study, we used the McDonald’s ω instead to evaluate the composite reliability, and it was acceptable for Factor 1 (ω = 0.80) and Factor 2 (ω = 0.87) with the total score (ω = 0.86), which is higher than 0.7, thus confirming the reliability of participants’ responses to the items of the scale.

Our results showed evidence of invariance by gender at the metric, configural, and scalar levels. It is of note that our study is the first to show the invariance between genders (not shown in the original study). Therefore, the measurement with the SSQ is appropriate for both male and female participants. This is a significant discovery because gender disparity is mostly seen as a problem in developing nations, where old sexist attitudes that minimize the value of women’s sexual satisfaction are still widely accepted [64].

Regarding the correlates of sexual satisfaction, our study was the first to highlight the negative relationship between waterpipe dependence and sexual satisfaction. This could be due to the high amounts of nicotine that cause more erectile dysfunction in men through arteriogenic and atherosclerotic mechanisms, for example, resulting in penile artery vasospasm [65]. It is also worth mentioning that nicotine has been noted to have a deleterious impact on testosterone production, which is the primary regulator of male sexual function [66]. Smoking might also impair ovarian function by reducing estrogen levels, and thus decreasing vaginal lubrication, which in its turn increases the risk of sexual dysfunction among females [67]. It is of note that the correlation between waterpipe smoking and sexual satisfaction is significant but small and explains little variance in sexual satisfaction. Therefore, our results should be interpreted with caution.

Our research findings also discovered a perplexing absence of connection between dependency on cigarettes and sexual satisfaction, in contrast to the observed negative correlation between smoking a waterpipe and sexual satisfaction. This inconsistency implies potential variations in the mechanisms underlying the impact of different forms of tobacco use on sexual well-being. Although cigarette smoking has long been associated with numerous adverse health outcomes [68], the distinct chemical composition and inhalation patterns related to waterpipe smoking may contribute to its differential effects on sexual satisfaction. In fact, the relatively long duration of a waterpipe use episode results in considerably greater effective nicotine exposure, combined with heightened levels of harmful substances such as carbon monoxide in waterpipe smoke, which may exert more pronounced physiological effects on general health than cigarettes [69], potentially causing decreased sexual function and satisfaction.

It is also noteworthy that in some Middle Eastern cultural settings, cannabis is occasionally mixed with tobacco in waterpipe smoking [70]. Considering that erectile dysfunction is twice as high in chronic cannabis users compared to controls [71], it is plausible that incorporating cannabis in waterpipe smoking rituals may have contributed to the observed negative correlation with sexual satisfaction within the Lebanese context.

Greater emotional intelligence remained associated with higher sexual satisfaction in both genders in the multivariable analysis, which is another important finding of the current study. This could be explained by the fact that these individuals have higher self-esteem, courage, and confidence, and subsequently know how to increase their emotional capacities in their marital life in order to experience satisfactory sexual relations [34].

Another noteworthy finding of this study is that physical activity was associated with higher levels of sexual satisfaction, a result which is consistent with previously reported findings. In fact, a recent study conducted among women found that physical activity may improve genital blood circulation by lowering clitoral vascular resistance [21]. Indeed, it has been proposed in another study that exercising may influence genital arousal as well by enhancing the activity of the sympathetic nervous system or the pelvic floor muscles [72]. Moreover, erectile function among men was found to improve after losing weight and increasing physical activity [73].

The findings of this investigation have significant clinical implications. Clinicians need to understand the variety of factors that contribute to sexual dissatisfaction in order to try to reduce their impact, especially considering the deleterious effects of waterpipe dependence, which could be prevented. Additionally, training workshops should be organized to focus on the significance of discussing sexual problems because normally Lebanese individuals ignore their sexual needs and concerns. Furthermore, it is recommended that education providers pay more attention to emotional intelligence, which is an acquired set of skills and behaviors [74], and thus focus on teaching different emotional intelligence skills. Also, from a clinical point of view, people should be encouraged to engage in sport activities and maintain a healthy BMI, not only for its beneficial effects on general health but also on sexual health.

There are several limitations within our study that warrant attention. Initially, our cross-sectional design limited our ability to establish causality. Additionally, the use of an online questionnaire might have constrained the generalizability of our findings and introduced selection bias, especially considering that the two samples are different. Moreover, low percentages of participants with high smoking dependence were found, for example, mainly due to the convenient sampling technique used to recruit participants. Moreover, potential response biases could have influenced the results, considering the sensitive nature of the subject matter and the possibility of participants experiencing shame or embarrassment. Similar to the potential impact of response bias, we did not implement measures to control for social desirability bias, which might have influenced the participants’ responses, particularly in the context of sensitive topics related to sexual health. Furthermore, as our sample was population-based, any assertions regarding the reliability of participants’ responses to the scale within clinical samples are not possible. Another limitation lies in the absence of other variables that could potentially correlate with sexual satisfaction. It is also important to note that our research did not include any evaluations made by external observers. This aspect could have provided a more comprehensive and unbiased assessment of the variables under investigation. Moreover, we did not thoroughly assess the potential interactions among the predictors, particularly their interactions with gender, which could have yielded valuable insights into differential effects based on gender-specific dynamics. Lastly, the relatively young age of responders is especially concerning, considering the fact that smoking is related to more severe health problems for those who smoke for a longer period of time [75]. Consequently, further investigations are imperative in order to address this particular issue within the older Lebanese population.

## 5. Conclusions

According to our present research, most of the factors identified to be related to sexual satisfaction are preventable. These findings suggest some practical implications for sexual satisfaction enhancement strategies in the Lebanese population through reducing waterpipe use along with boosting emotional intelligence and increasing physical activity. The reliability and validity of the responses to the Arabic version of the Sexual Satisfaction Questionnaire are likewise supported by our findings. Through this scale, we encourage cross-cultural research on sexual satisfaction among Arabic-speaking people in a variety of settings. However, more research is required to understand the role of the new trending smoking types, such as electronic cigarettes, IQOS, and vapes, in sexual health as they were not included in our study.

## Figures and Tables

**Table 1 healthcare-11-03068-t001:** Participants.

Study 1	Study 2
➢ 270 Participants ➢ Mean age M = 33.68 (SD = 3.83) ➢ 135 females (50.2%) ➢ 147 single (54.6%) ➢ 185 with a university level of education (68.8%)	➢ 359 Participants ➢ Mean age M = 22.67 (SD = 5.60) ➢ 228 females (63.5%) ➢ 314 single (87.5%) ➢ 337 with a university level of education (93.9%)

**Table 2 healthcare-11-03068-t002:** Items of the SSQ in English, factor loadings from the first sample’s exploratory factor analyses (EFA), and standardized estimates of factor loadings from the second sample’s confirmatory factor analyses (CFA).

	EFA	CFA
Item		
Factor 1: Positive aspect of sexual satisfaction (positively-worded items)		
2. “Sex is a source of pleasure for me”	0.94	0.82
4. “I feel sexually attractive”	0.98	0.77
6. “I do not have any problems in my sexual life”	0.69	0.41
7. “I like thinking about my sexual life”	0.81	0.88
10. “I find my sexual life fulfilling”	0.69	0.38
Factor 2: Negative aspect of sexual satisfaction (negatively-worded items) *		
1. “I am disconcerted with a part of my sexual life”	0.86	0.73
3. “Thinking about sex generates negative emotions”	0.80	0.59
5. “I find myself a poor sexual partner”	0.97	0.81
8. “My sexual life frustrates me”	0.86	0.82
9. “I am afraid I do not satisfy my sexual partner”	0.81	0.81

* Items of Factor 2 were reversed.

**Table 3 healthcare-11-03068-t003:** Measurement invariance across gender in the second sample.

Model	χ^2^	df	CFI	RMSEA	SRMR	Model Comparison	Δχ^2^	Δdf	*p*	ΔCFI	ΔRMSEA	ΔSRMR
Configural	200.54	66	0.919	0.076	0.085							
Metric	214.44	74	0.916	0.073	0.087	Configural vs. metric	13.9	8	0.084	0.003	0.003	0.002
Scalar	218.27	82	0.918	0.068	0.087	Metric vs. scalar	3.83	8	0.872	0.002	0.005	<0.001

Note: CFI = comparative fit index; RMSEA = Steiger–Lind root-mean-square error of approximation; SRMR = standardized root-mean-square residual.

**Table 4 healthcare-11-03068-t004:** Correlations of the SSQ score with the other measures in the second sample.

	1	2	3	4	5	6	7	8	9	10	11
1. Sexual satisfaction	1										
2. Cigarette dependence	−0.03	1									
3. Waterpipe dependence	−0.22 ***	0.03	1								
4. Alcohol use disorder	−0.11 *	0.31 ***	0.08	1							
5. Psychological distress	−0.17 **	−0.01	0.18 **	0.02	1						
6. Emotional intelligence	0.31 ***	−0.10	−0.08	−0.10	−0.07	1					
7. Age	0.12 *	0.27 ***	0.08	0.10	−0.02	0.04	1				
8. Body Mass index	0.01	0.18 ***	0.01	0.07	0.09	0.01	0.26 ***	1			
9. Household crowding index	−0.03	−0.03	−0.09	−0.04	−0.03	0.002	−0.15 **	−0.002	1		
10. Physical activity index	0.20 ***	0.07	−0.10	0.05	−0.07	0.01	−0.08	0.01	−0.06	1	
11. Financial burden	−0.05	0.11 *	0.09	0.05	0.30 ***	−0.04	0.10	0.11 *	0.10	−0.002	1

* *p* < 0.05; ** *p* < 0.01; *** *p* < 0.001.

**Table 5 healthcare-11-03068-t005:** Multivariate analysis: linear regression using the sexual satisfaction score as the dependent variable.

Variable	Unstandardized Beta	Standardized Beta	*p*	95% CI
Waterpipe dependence	−0.09	−0.17	**0.001**	−0.14; −0.04
Alcohol use disorder	−0.10	−0.09	0.078	−0.20; 0.01
Psychological distress	−0.06	−0.09	0.059	−0.12; 0.002
Emotional intelligence	0.15	0.27	**<0.001**	0.10; 0.20
Physical activity	0.03	0.17	**0.001**	0.02; 0.05
Marital status (married vs. single *)	1.36	0.11	0.072	−0.12; 2.83
Education level (university vs. secondary or less *)	1.15	0.07	0.158	−0.45; 2.75
Age	0.07	0.10	0.096	−0.01; 0.15

* Reference group; Nagelkerke R^2^ = 0.215; numbers in bold indicate significant *p* values.

## Data Availability

The data generated or analyzed during this study are not publicly available due to restrictions from the ethics committee. Reasonable requests can be addressed to the corresponding author.

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
