# Peer review of "Sexual Satisfaction among Lebanese Adults: Scale Validation in Arabic and Correlates Following Two Cross-Sectional Studies"

_healthcare, 2023, doi:10.3390/healthcare11233068_

Round 1
Reviewer 1 Report
Comments and Suggestions for Authors
-You only mentioned 'waterpipe' in the title, but that's not how the article went.
-The introduction should be shortened.
-Indicate the limitations with a fluid expression, not counting.
-The discussion part seems unsatisfactory.
Author Response
-You only mentioned 'waterpipe' in the title, but that's not how the article went.
The title was adjusted so that it mentions the correlates in general and not just “waterpipe dependence”.
-The introduction should be shortened.
Unfortunately, we were limited on that point since many factors needed to be introduced. Also, the study includes a scale validation part that we had to talk about. Some of the other reviewers even asked to add more studies and information. However, we still managed to remove some sentences.
-Indicate the limitations with a fluid expression, not counting.
The structure of this section was adjusted to be more fluid.
-The discussion part seems unsatisfactory.
Changes were made in this section, but could you please provide us with more details on what exactly did you find unsatisfactory so we can gladly work on it?
Reviewer 2 Report
Comments and Suggestions for Authors
Thank you very much for allowing me to review this very interesting article. Below I add some suggestions for improving the article.
The title could be summarized to validation which is the main objective.
The summary is not clear especially how many participants are there, is it a reliable instrument and the objective is not clear.
As for the introduction it is necessary to strengthen the theoretical framework with more current references and support it in theory and justification of the instrument with other scales and its necessity. Improve the objectives and no one hypothesis to be confirmed.
Summarize and organize the methodology. Design, participants, instruments, procedure and data analysis.
The results are clear and adequate.
The discussion should be a complete text and not in subsections.
Update to more recent references.
Author Response
Thank you very much for allowing me to review this very interesting article. Below I add some suggestions for improving the article.
The title could be summarized to validation which is the main objective.
We agree, and the title was adjusted to include the validation but with the correlates in general which are also a main part of this study.
The summary is not clear especially how many participants are there, is it a reliable instrument and the objective is not clear.
- Participants: In the summary we mentioned that two cross-sectional studies were conducted between June and September 2022 with 270 and 359 participants respectively.
- Objectives: Main objective à assess the reliability and validity of the Arabic version of the Sexual Satisfaction Questionnaire. Secondary objective à investigate the correlates of sexual satisfaction.
- The conclusion part in the summary was adjusted to include that the SSQ was found to be a reliable instrument.
As for the introduction it is necessary to strengthen the theoretical framework with more current references and support it in theory and justification of the instrument with other scales and its necessity. Improve the objectives and no one hypothesis to be confirmed.
The objectives were improved and a hypothesis was added. More recent references were added to strengthen the theoretical framework. A more detailed justification of the necessity to have an Arabic version of this Questionnaire was added.
Summarize and organize the methodology. Design, participants, instruments, procedure and data analysis.
The methodology was organized as requested. We tried to summarize it as much as possible without losing important information.
The results are clear and adequate.
Thank you we really appreciate it.
The discussion should be a complete text and not in subsections.
This section was adjusted to be a complete fluid text.
Update to more recent references.
More recent references were added.
Reviewer 3 Report
Comments and Suggestions for Authors
Moderate editing of English language required
Author Response
Moderate editing of English language required
The manuscript was reviewed for English language.
In this article authors aimed to evaluate the psychometric properties of the Arabic version of the Sexual Satisfaction Scale among a sample of Lebanese adults, through the analysis of two cross-sectional studies results. They found that most of the factors related to sexual satisfaction identified are preventable as the waterpipe dependence. Even if the topic is interesting, results need to be better argued because they are sometimes difficult to understand and so are the tables.
At the moment, the statistical part is predominant so perhaps it would be more suitable for another type of journal.
We understand the importance of ensuring the alignment between our research and the aims and scope of the journal. However, we believe that our findings have significant implications for the readership of Healthcare, particularly in the context of sexual health which constitutes an important part of everyone’s general well-being. Also, this study isn’t only about statistics as it has further perspectives to shed the light on a sensitive topic and be a reference for future prevention campaigns and so on…
Regarding the tables, we tried to include clear titles, with appropriate footnotes for better clarity to the reader. We appreciate it if you could point out the things that are not clear so we tackle them. Thank you for your understanding.
Minor points: I think that should be specified in the title that the article is based on two cross-sectional studies and the correlation with waterpipe dependence could be omitted not being the only one investigated.
We omitted “waterpipe dependence” and added that the study is based on two cross-sectional studies.
Why did you decide to carry out two separate cross-sectional studies? Please add the motivations in the text.
We followed the appropriate designs for study validations: one sample was used for the EFA and the second one for the CFA. It is not about motivation but it is about following international recommendations (please refer to Swami and Barren, 2019).
Lines 144-148 please insert participants’ information in a table to make them clearer. References should be revised because some of them are too old and other missed information (lines 514 and 533).
Participants’ information were put in a table.
We updated to newer references, and the ones missing information were corrected.
Reviewer 4 Report
Comments and Suggestions for Authors
This is a very interesting work done in a society where people are not used to discussing issues related to sexual experiences. Its aim was “to evaluate the psychometric features of the Sexual Satisfaction Questionnaire as well as the correlates of sexual satisfaction, especially waterpipe dependence, in a population of Lebanese adults”. The authors have used sophisticated statistical tests to analyze their data and the paper is suitable for your International Journal. In order to improve the understanding of this manuscript, I have a few suggestions for the authors.
Introduction
Although there is a paragraph about physicians’ efforts to discuss sexual matters with their patients, I feel that because of this limited information the authors could, perhaps, provide some additional relevant information and possibly references to the status and relationship of sexual health in the country, especially to young people. Emphasis must be placed on the scientific gap in Lebanese society that this study aspires to fill.
2. Materials and methods
2.1. Procedures and Participants
I'm not sure I understand the use of the 2 studies, could you give some explanation, please.
The sample composition is so different in each study, is there a specific reason or is it because this is a convenience sample?
2.3. Measures
It is suggested to give examples of some relevant items that each scale contains e.g. SSS, LWDS-11, SDQ etc.
I am curious if all the scale constructs used are non-homogeneous, or if there were differences in reliability between items that forced you to use the Omega instead of the α test.
2.5. Statistical analyses
It is only right that you have done both exploratory and confirmatory Factor Analysis! I would also like you to name each factor according to the items it contains.
Author Response
This is a very interesting work done in a society where people are not used to discussing issues related to sexual experiences. Its aim was “to evaluate the psychometric features of the Sexual Satisfaction Questionnaire as well as the correlates of sexual satisfaction, especially waterpipe dependence, in a population of Lebanese adults”. The authors have used sophisticated statistical tests to analyze their data and the paper is suitable for your International Journal. In order to improve the understanding of this manuscript, I have a few suggestions for the authors.
Introduction
Although there is a paragraph about physicians’ efforts to discuss sexual matters with their patients, I feel that because of this limited information the authors could, perhaps, provide some additional relevant information and possibly references to the status and relationship of sexual health in the country, especially to young people. Emphasis must be placed on the scientific gap in Lebanese society that this study aspires to fill.
More information and references in order to emphasis this idea were added
- Materials and methods
2.1. Procedures and Participants
I'm not sure I understand the use of the 2 studies, could you give some explanation, please.
We followed the appropriate designs for study validations: one sample was used for the EFA and the second one for the CFA. We followed international recommendations (please refer to Swami and Barren, 2019).
The sample composition is so different in each study, is there a specific reason or is it because this is a convenience sample?
It is probably related to convenient sampling. We added your idea as a limitation as follows:
Additionally, the use of an online questionnaire might have constrained the generalizability of our findings and introduced selection bias, especially that the two samples are different.
2.3. Measures
It is suggested to give examples of some relevant items that each scale contains e.g. SSS, LWDS-11, SDQ etc.
Some examples of items were added to the mentioned scales.
I am curious if all the scale constructs used are non-homogeneous, or if there were differences in reliability between items that forced you to use the Omega instead of the α test.
McDonald’s ω was selected as a measure of composite reliability because of known problems with the use of Cronbach’s α.
Reference: McNeish D. Thanks coefficient alpha, we’ll take it from here. Psychol Methods. 2018;23(3):412. doi: 10.1037/met0000144.
2.5. Statistical analyses
It is only right that you have done both exploratory and confirmatory Factor Analysis! I would also like you to name each factor according to the items it contains.
We named the factors as requested (please refer to Table 2).
Reviewer 5 Report
Comments and Suggestions for Authors This paper is based on a useful design, with acceptable psychometric analysis, including back translation of instruments. The method of obtaining subjects is novel, and ingenious. "Water Pipe" use should be removed from the title, since this could be confused with drinking water supply (a measure of poverty). Water Pipe smoking is the strongest correlate of poorer sexual adjustment. Nevertheless, the correlation although significant is small, and explains very little variance in the dependent variable (sexual adjustment). This should be made clear. This correlation is by no means strong, and further studies are required in order to explain sexual satisfaction or adjustment. It is not clear why the data from the second study were not included in the overall analysis. McDonald's Omega needs an up-to-date reference. It is known that in some cultures, cannabis is added to tobacco in water pipe smoking. There is some evidence that chronic use of cannabis can cause erectile dysfunction. Would the authors care to comment or speculate on this possibilty in explaining effects of water pipe smoking in their own culture?Comments on the Quality of English Language
Further English language editing is necessary.
Author Response
"This paper is based on a useful design, with acceptable psychometric analysis, including back translation of instruments. The method of obtaining subjects is novel, and ingenious.
We would like to thank you for your appreciation J
"Water Pipe" use should be removed from the title, since this could be confused with drinking water supply (a measure of poverty).
Removed from the title.
Water Pipe smoking is the strongest correlate of poorer sexual adjustment. Nevertheless, the correlation although significant is small, and explains very little variance in the dependent variable (sexual adjustment). This should be made clear. This correlation is by no means strong, and further studies are required in order to explain sexual satisfaction or adjustment.
The term “strong” was removed from the paper.
We added your idea to the discussion section as follows:
Regarding the correlates of sexual satisfaction, our study was the first to highlight the negative relation between waterpipe dependence and sexual satisfaction. This could be due to the high amounts of nicotine that cause more erectile dysfunction in men through arteriogenic and atherosclerotic mechanisms for example, resulting in penile artery vasospasm (Verze et al., 2015). It is also worth mentioning, that nicotine has been noted to have a deleterious impact on testosterone production, which is the primary regulator of male sexual function (Park et al., 2012). Smoking might also impair the ovarian function by reducing estrogen levels, and thus decreases the vaginal lubrication, which in its turn increases the risk of sexual dysfunction among females (Choi et al., 2015). Although waterpipe smoking is the strongest correlate of poorer sexual adjustment, nevertheless, the correlation although significant is small, and explains little variance in the dependent variable (sexual adjustment). Therefore, our results should be interpreted with caution.
It is not clear why the data from the second study were not included in the overall analysis.
The variables associated with sexual satisfaction were only included in the second sample. They were not included in the first one.
McDonald's Omega needs an up-to-date reference.
Reference updated to McNeish 2018.
It is known that in some cultures, cannabis is added to tobacco in water pipe smoking. There is some evidence that chronic use of cannabis can cause erectile dysfunction. Would the authors care to comment or speculate on this possibilty in explaining effects of water pipe smoking in their own culture?
Thank you for pointing out this idea. It is really possible that the cannabis inside the waterpipe to have a possible negative correlation with sexual satisfaction. We added an additional comment to the discussion of waterpipe dependence.
Reviewer 6 Report
Comments and Suggestions for Authors
See attached file

Some English awkwardness throughout the manuscript.
Round 2
Reviewer 1 Report
Comments and Suggestions for Authors
Accept
Author Response
Thank you
Reviewer 2 Report
Comments and Suggestions for Authors
Thank you for your review. I consider that the authors have answers the questions suggested. I would advise of a short title.
Author Response
Thank you for your appreciation. We tried to remove two words out of the title as per your request.
Reviewer 3 Report
Comments and Suggestions for Authors
The manuscript has been sufficiently improved to warrant publication in Healthcare.
Comments on the Quality of English LanguageMinor editing of English language required
Author Response
The paper was edited for English language one more time. Thank you.
Reviewer 5 Report
Comments and Suggestions for Authors
The authors have satisfactorily addressed the comments of Reviewers, and I now recommend publication of this paper.
Author Response
Thank you.
Reviewer 6 Report
Comments and Suggestions for Authors
See attached file

Needs to have an English language editor.
